# Prostate Cancer Segmentation using Manifold Mixup U-Net

**Wonmo Jung**[1]                                                                    WM.JUNG@VUNO.CO

**Sejin Park**[1]                                                                     GNOSES@VUNO.CO

**Kyu-Hwan Jung**[1]                                                              KHWAN.JUNG@VUNO.CO

**Sung Il Hwang**[2]                                                             HWANGSI49@GMAIL.COM

[1] *VUNO Inc, Seoul, South Korea*

[2] *Department of Radiology, Seoul National University Bundang Hospital, Seongnam, South Korea*

**Editors:** Under Review for MIDL 2019

## Abstract

The scarcity of labeled data is a challenging problem in medical segmentation. Here, we suggest to apply manifold mixup, a recently proposed simple regularizer that utilizes linear combinations of hidden representations of training examples, on prostate cancer segmentation using MR image. Manifold mixup applied to either the encoder or decoder outperformed training without mixup and mixup applied on the input space.

**Keywords:** Regularization, Generalization, Data Augmentation, Mixup, Image Segmentation

## 1. Introduction

In medical image segmentation, the scarcity of labeled data is a well-known problem related to generalizability. With the constrained availability of labeled training data, deep neural networks often provide incorrect but confident predictions on test samples which are slightly different from training data. To tackle this problem, various approaches including data augmentation and semi- supervised learning, have been exploited. Recently, manifold mixup, a simple regularizer has been proposed to address this issue by training neural networks on linear combinations of hidden representations of training examples (Verma et al., 2018). This simple strategy enables neural networks to obtain smoother decision boundaries at multiple levels of representation and to obtain improved generalizability.

In this work, we apply manifold mixup to prostate MR images for the purpose of cancer segmentation. Specifically, we compare the effect of adopting manifold mixup to different parts of U-Net architecture, which has been widely used in medical image segmentation(Ronneberger et al., 2015). The parts that we applied the manifold mixup were the 1)encoder, 2)decoder, 3)skip connection, and 4)bottleneck of U-net architecture and the corresponding segmentation performances were obtained. Encoder and decoder mixups improved segmentation of prostate cancer (PCa) on multiparametric MR data while skip connection and bottleneck mixups did not show noticeable improvement. Applying manifold mixup on U-Net seems like a feasible approach to improve segmentation results with a simple modification of the pre-existing training strategy.

## 2. Materials and Methods

From 2011 Jan to 2018 Apr, 350 patients who underwent prostatectomy for PCa were enrolled retrospectively. Histopathologically confirmed ground truth label of PCa were drawn on the T2 weighted image by a uroradiologist with 19 years of experience. Among 350 MR images, randomly selected 50 cases were used as the test set.

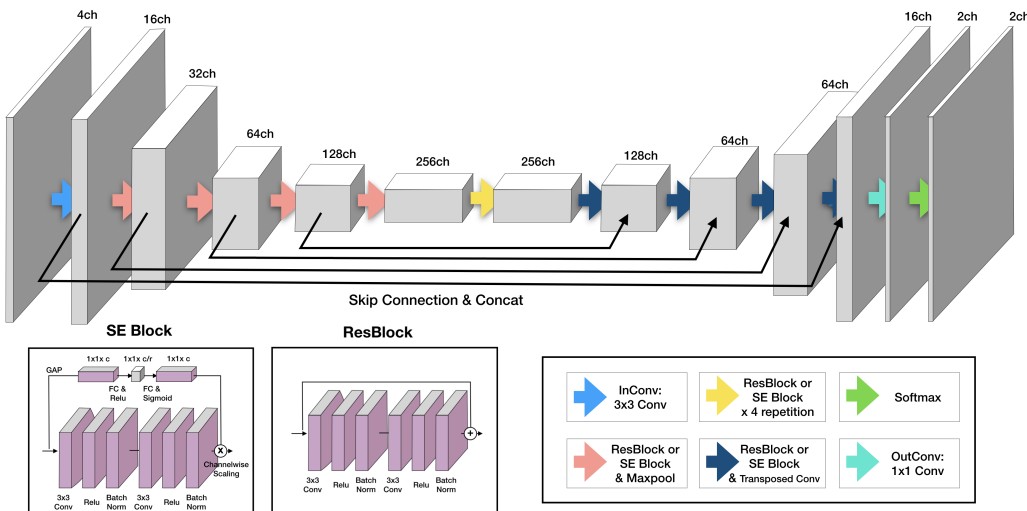

Figure 1: Modified U-net architecture with SE block or ResBlock for PCa segmentation.

Our network is a variation from the U-Net architecture containing three modules: encoder, bottleneck, and decoder. Two versions of the convolution blocks, SE block utilizing squeeze and excitation(Hu et al., 2017) and ResBlock which has a short residual connection inside the block(He et al., 2015), were tested as shown in Figure 1.

Four types of mixup architectures are at their respective locations of the mixup operation. The mixup location for each type is specified in Figure 2. The mixup operation of each location is randomly performed with a binomial probability of 0.5. Since more than two mixup operations can be performed during one batch training, the mixup sequence was recorded to be used for the GT label mixup. A mixup operation is a linear combination: $h_{mixup} = \lambda(h) + (1 - \lambda)(h')$, $y_{mixup} = \lambda(y) + (1 - \lambda)(y')$ of two training data: $(x, y)$ and $(x', y')$, where $h$ and $h'$ are hidden representation of $x$ and $x'$. The parameter $\lambda \in [0, 1]$ is distributed according to a Beta distribution: $\lambda \sim \beta(\alpha, \alpha)$. In this study, an $\alpha$ of 2.0 was used. The pair of training data was randomly selected in a minibatch through permutation.

Manifold mixup has been used on image classification problems and generative adversarial networks (Verma et al., 2018), but not in semantic segmentation. Also, input mixup which applies linear combinations in the input space has been successfully applied for brain tumor segmentation(Eaton-Rosen and Cardoso, 2018), but not manifold mixup. Therefore, we first tested whether the input mixup benefits PCa segmentation. Then, we evaluated our four different types of manifold mixups. The network was trained to predict PCa lesions using Dice loss and the performance was also evaluated using the Dice coefficient between expert annotations and the output of the segmentation network. We used the SGD optimizer with an initial learning rate of 0.05 with momentum 0.9 and weight decay $10^{-4}$ for

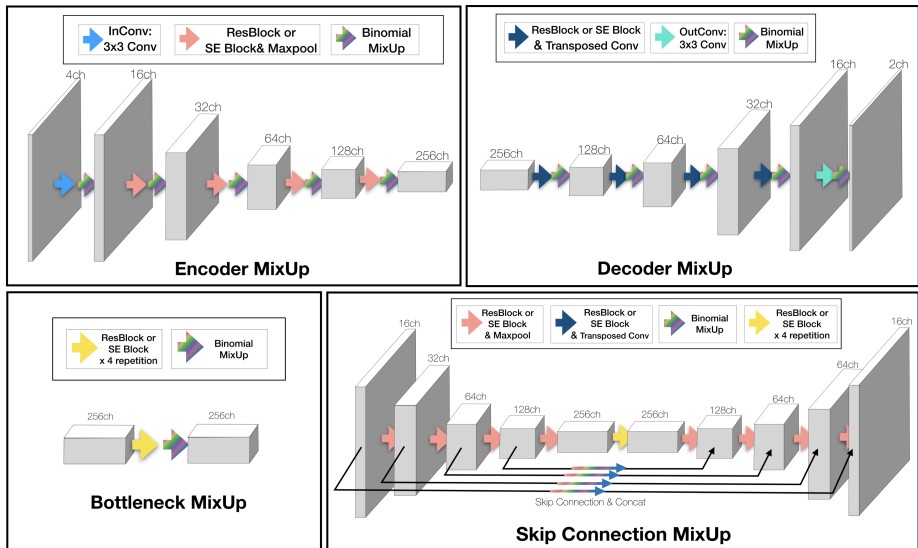

Figure 2: Applications of mixup to different parts of U-net architecture : Encoder, decoder, bottleneck and skip connection.

all experiments. The training was terminated if the moving average of validation Dice loss did not improve by more than $1 \times 10^{-3}$ within the last 20 epochs.

## 3. Results

We demonstrated the segmentation performance of the various methods in Table 1. The best Dice coefficient was achieved through SE-block U-Net with a manifold mixup in the decoder module. While the input mixup did not improve training without a mixup, two (encoder and decoder) out of our four manifold mixup types outperformed training without a mixup.

Table 1: Result of different applications of mixup to PCa segmentation task

| Mixup Types | U-Net(ResBlock) | U-Net(SE-Block) |
|---|---|---|
| Without Mixup | 0.4469 | 0.4606 |
| Input Mixup | 0.4610 | 0.4306 |
| Encoder Mixup | 0.4774 | 0.4816 |
| Bottleneck Mixup | 0.4302 | 0.4288 |
| Skip Connection Mixup | 0.4645 | 0.4320 |
| Decoder Mixup | **0.4826** | **0.4836** |

## 4. Conclusion

We applied manifold mixup which uses linear combination of hidden representations of randomly selected two samples on segmentation of PCa using multiparametric MR image. The results demonstrated that the mixup of hidden representations in the encoder and decoder part of U-net improved segmentation performance.

## Acknowledgments

This work was supported by Institute for Information communications Technology Promotion(IITP) grant funded by the Korea government(MSIT)(No.2018-0-00861, Intelligent SW Technology Development for Medical Data Analysis)

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
