# OpenReview forum: "Prostate Cancer Segmentation using Manifold Mixup U-Net"
_MIDL.io/2019/Conference/Abstract — MIDL Abstract 2019_

### Official Review · AnonReviewer1 · 2019-04-29

**Rating:** 3
**Confidence:** 3

**Review:**

This work applies manifold mix-up for prostate segmentation. The topic is worthy study and valuable in the medical imaging community with limited dataset available. Experimental design and validations are sound.
Cons: The absolute value of segmentation performance is still not very high.

---

### Official Review · AnonReviewer2 · 2019-05-01
**Application of manifold mixup for prostate cancer segmentation in MRI.**

**Rating:** 3
**Confidence:** 3

**Review:**

This paper presents an interesting application of the recently proposed manifold mixup regularization technique to a U-Net architecture for prostate image segmentation in MR images. Manifold mixup was applied to different parts of the networks, and the authors reported improvement with respect to the baseline model for 2 out of 4 configurations. The paper is easy to follow and clearly explained.

My only concern with this work is the overall low Dice coefficient reported for the segmentation task. I would have liked to see some figures with the MR images, ground truth and predictions, to understand why the task is so difficult.

---

### Decision · Program_Chairs · 2019-05-06
**Acceptance Decision**

Accept